# Impact of Contextual-Level Social Determinants of Health on Newer Antidiabetic Drug Adoption in Patients with Type 2 Diabetes

**DOI:** 10.3390/ijerph20054036

**Published:** 2023-02-24

**Authors:** Yujia Li, Hui Hu, Yi Zheng, William Troy Donahoo, Yi Guo, Jie Xu, Wei-Han Chen, Ning Liu, Elisabeth A. Shenkman, Jiang Bian, Jingchuan Guo

**Affiliations:** 1Department of Pharmaceutical Outcomes & Policy, College of Pharmacy, University of Florida, Gainesville, FL 32610, USA; 2Channing Division of Network Medicine, Brigham and Women’s Hospital, Boston, MA 02115, USA; 3Department of Medicine, Harvard Medical School, Boston, MA 02115, USA; 4Division of Endocrinology, Diabetes and Metabolism, College of Medicine, University of Florida, Gainesville, FL 32610, USA; 5Department of Health Outcomes & Biomedical Informatics, College of Medicine, University of Florida, Gainesville, FL 32610, USA

**Keywords:** social determinants of health, type 2 diabetes, antidiabetic drugs, neighborhood deprivation index, vacant land

## Abstract

Background: We aimed to investigate the association between contextual-level social determinants of health (SDoH) and the use of novel antidiabetic drugs (ADD), including sodium-glucose cotransporter-2 inhibitors (SGLT2i) and glucagon-like peptide-1 receptor agonists (GLP1a) for patients with type 2 diabetes (T2D), and whether the association varies across racial and ethnic groups. Methods: Using electronic health records from the OneFlorida+ network, we assembled a cohort of T2D patients who initiated a second-line ADD in 2015–2020. A set of 81 contextual-level SDoH documenting social and built environment were spatiotemporally linked to individuals based on their residential histories. We assessed the association between the contextual-level SDoH and initiation of SGTL2i/GLP1a and determined their effects across racial groups, adjusting for clinical factors. Results: Of 28,874 individuals, 61% were women, and the mean age was 58 (±15) years. Two contextual-level SDoH factors identified as significantly associated with SGLT2i/GLP1a use were neighborhood deprivation index (odds ratio [OR] 0.87, 95% confidence interval [CI] 0.81–0.94) and the percent of vacant addresses in the neighborhood (OR 0.91, 95% CI 0.85–0.98). Patients living in such neighborhoods are less likely to be prescribed with newer ADD. There was no interaction between race-ethnicity and SDoH on the use of newer ADD. However, in the overall cohort, the non-Hispanic Black individuals were less likely to use newer ADD than the non-Hispanic White individuals (OR 0.82, 95% CI 0.76–0.88). Conclusion: Using a data-driven approach, we identified the key contextual-level SDoH factors associated with not following evidence-based treatment of T2D. Further investigations are needed to examine the mechanisms underlying these associations.

## 1. Introduction

More than 100,000 individuals die from diabetes each year in the United States (US) [1]. Of these deaths, 60% are attributed to concurrent cardiovascular disease (CVD), with myocardial infarction being the most common cause [2]. Among the antidiabetic drugs (ADD) currently available on the US market, two relatively novel agents, sodium-glucose cotransporter-2 inhibitors (SGLT2i) and glucagon-like peptide-1 receptor agonists (GLP1a), are associated with significant reductions in blood glucose levels and have been found particularly effective in reducing the risk of CVD in individuals with type 2 diabetes (T2D) [3]. In addition, these novel antidiabetic agents have been shown to associate with weight loss, reduced risk of hypoglycemia and cardiorenal protection, favorable benefits that are of great importance to patients with T2D [3]. The American Diabetes Association (ADA) recommends SGLT2i and GLP1a for patients with T2D who have CVD, heart failure, chronic kidney disease, or an increased risk of these conditions, regardless of their glycemic status [4,5].

However, the utilization of SGLT2i and GLP1a in real-world T2D patient populations is relatively low in the US compared to other ADD [6,7], especially among historically marginalized communities, such as racial and ethnic minority groups and individuals experiencing socioeconomic disadvantages. Data from commercial insurance and Medicare, for example, showed that Black patients were 10–20% less likely to receive newer ADD than White patients [7,8,9,10]. While such disparity can be explained overall by racial disparity as a distal cause, its proximal cause—the underlying mechanism whereby racial and ethnic groups have initiated SGLT2i/GLP1a—remains largely unknown. 

In the past, research and clinical approaches centered on the individual-level have led improvements in self-management outcomes and reduction in cardiovascular risk among patients with T2D [11]. More recently, researchers have acknowledged the need to consider external factors, namely the social determinants of health (SDoH) to achieve the goal of sustainable improvement in diabetes outcomes [12]. SDoH refer to the various social, economic, and environmental factors, including access to healthcare, education, employment, housing, and social support that have an impact on people’s health, well-being, and quality of life [13]. Contextual-level SDoH refers to the broader social and built factors within community or region that influence health outcomes, and are increasingly recognized as a vital source of information to develop healthcare policies designed to improve population health management and value-based care [14,15]. Previous studies have demonstrated the association of contextual-level SDoH with geographic variation and diabetes risk [16]. However, minimal data exist on the extent to which contextual-level SDoH (e.g., residential segregation, food environment, and neighborhood walkability) may impact healthcare use, including initiating evidence-based treatment in T2D care [8]. A Dutch study published in 2012 examined the association of regional-level aging composition and socioeconomic status with spatial variation in ADD use but without a comprehensive evaluation of multiple contextual-level SDoH [17]. 

Given that race and ethnicity are social constructs [18], contextual-level SDoH can play important roles in the development of racial and ethnic disparities across geographic regions [19]. Therefore, understanding how contextual-level SDoH impact the adoption of these outcome-improving therapies in millions of Americans with T2D is imperative. Accordingly, this study aimed to examine the association between patients’ contextual-level SDoH and their initiation of the newer ADD, and how such associations may vary across racial and ethnic groups. With such empirical evidence, the racial disparity in SGLT2i/GLP1a utilization can be better understood, and relevant policymaking can be better guided. 

## 2. Materials and Methods

### 2.1. Data Source and Study Population

This is a retrospective cohort study using data from the OneFlorida+ network, containing large collections of electronic health records (EHR) covering more than 19 million patients from Florida (~16.8 million), Georgia (~2.1 million), and Alabama (~9.1 thousand) [20]. We assembled a cohort of adults (i.e., aged ≥ 18) identified as having at least one inpatient or outpatient T2D diagnosis (using ICD-9 codes 250.x0 or 250.x2, or ICD-10 code E11) and ≥1 ADD prescription. The algorithm used to identify T2D has been validated in OneFlorida+ with a positive predictive value (PPV) > 94 [21] and is preferred over using only diagnosis codes, which can lead to misclassification error [22]. Among the T2D cohort, we identified individuals who initiated SGLT2i or GLP1a, or another second-line ADD (i.e., dipeptidyl-peptidase-4 inhibitors, sulfonylureas, thiazolidinediones, and basal insulin) in 2015–2020. The index date was the day of the first prescription of a second line ADD, defined as no use of the drug in three prior years. We restricted the study cohort by only including those individuals who had ≥ 2 inpatient or outpatient encounters per year in OneFlorida+ in the three years prior to the index date to obtain complete information for modelling.

### 2.2. Study Outcome and Covariates

The outcome was the initiation of a newer ADD (i.e., SGLT2i or GLP1a) versus another second-line drug. We collected baseline demographic and clinical information on or within the 3-year period prior to the index date, including age, sex, race-ethnicity (non-Hispanic White [NHW], non-Hispanic Black [NHB], Hispanic, and other), rurality (defined using linkage to rural–urban continuum codes [RUCC] based on patients’ residencies’ Federal Information Processing System [FIPS] county code and classified the rurality into three levels by the US Department of Agriculture’s (USDA) Economic Research Service: RUCC ≤ 3 as metropolitan; 3 < RUCC ≤ 7 as urban; and 7 < RUCC ≤ 9 as rural), primary payer (Medicare, Medicaid, private insurance, no insurance, and other), diabetes complications and comorbidities (such as cardiovascular disease and chronic kidney disease), co-medications (i.e., use of another ADD, antihypertensives, statins, and antidepressants), clinical presentation (most recent blood pressure and body mass index [BMI], identified in four categories: ≤25, 25–30, 30–100 kg/m^2^, or missing), and lab values (most recent hemoglobin A1c [HbA1c], identified in four categories: ≤7, 7–10, 10–21 mmHg, or missing). Clinical data were extracted from de-identified EHR records in the OneFlorida+ network.

### 2.3. Contextual-Level SDoH 

We obtained data on built and social environment measures from six well-validated sources with different spatiotemporal scales, characterizing food access, walkability, vacant land, neighborhood disadvantage, social capital, crime and safety. All measures were spatiotemporally linked to each individual considering residential mobility during the study period. Area-weighted averages were first calculated according to a 250 m buffer around the centroid of each 9-digit ZIP code. Time-weighted averages were then calculated, accounting for each individual’s residential history.

Table 1 summarizes the contextual-level data sources and the corresponding spatiotemporal scales. A total of 43 food access measures at census tract level in 2015 and 2019 were obtained from USDA’s Food Access Research Atlas [23]. Walkability was assessed using the National Walkability Index developed by the US Environmental Protection Agency (EPA) [24], which assesses walkability on a scale from 1 to 20 for each census block group, with 1 indicating the least walkable and 20 the most walkable. Vacant land measures at the census-tract level from 2015 to 2019 were obtained from the US Department of Housing and Urban Development aggregated with US Postal Service administrative data [25] and a total of 18 measures that were available across all years were included. The neighborhood deprivation index (NDI), a socioeconomic status measure, was obtained at the census block group level based on data from the 2015 to 2019 American Community Survey (ACS). It yields information on the income, education, employment, and housing quality of a neighborhood and allows ranking by socioeconomic disadvantage [26]. In addition, ten social capital measures were constructed using the Census Business Pattern data based on the North American Industry Classification System (NAICS) codes [27] at the 5-digit ZIP code tabulation area (ZCTA5) level. Furthermore, eight county-level annual measures of crime and safety were obtained from the Uniform Crime Reporting Program from 2015 to 2019 [28]. A total of 81 SDoH measures were included in the analyses.

### 2.4. Statistical Analysis 

We conducted normalization transformations for all continuous contextual-level SDoH variables using the *bestNormalize* package in R, which implements several transformation methods, including log, square root, exponential, arcsinh, box cox, and Yeo-Johnson transformations [29]. The best transformation was determined based on Pearson *P* statistics. All continuous variables were also z-score standardized (mean = 0 and standard deviation = 1). All contextual-level SDoH factors and covariates of interest described above had missing values for <2% of the participants; Missing data for all contextual-level SDoH factors were imputed using the chained equations method of the MICE package in R. A variable was considered a predictor in the imputation model if its proportion of non-missing values among counties with missing values in the variable to be imputed was larger than 40% and they were correlated (i.e., with the absolute correlation value > 0.4) with the variable to be imputed or the probability of the variable being missing. We imputed a single dataset given the minimal impacts of the imputation procedure due to the large sample size and a small fraction of missing data. Missing information on BMI and HbA1c were not imputed and maintained as a separate category.

We used a two-phase approach to identify key contextual-level associated with initiation of SGTL2i/GLP1a versus other second-line ADD [30,31]. In Phase 1, we randomly split the data into a 50% discovery set and a 50% replication set. We considered all the 83 contextual-level SDoH for associations with newer ADD initiation after accounting for multiple comparisons. We built multivariable logistic regression models for each contextual-level factor after adjusting for demographics, urbanicity, diabetes complications, co-medications, clinical presentation, and primary payer. To account for the multiple testing, the Benjamin-Hochberg procedure was used to control the false discovery rate (FDR) at 5% [32]. A variable was considered significant if it had an FDR-adjusted *p*-value (or *q*-value) < 0.05 in both the discovery and the replication sets. A correlation heatmap was generated to show the pairwise Pearson correlations of the variables retained from Phase 1. Variables from highly correlated pairs (with the absolute value of correlation coefficients > 0.6) were removed to avoid collinearity between variables [33]. 

In Phase 2, we used a multivariable logistic regression model including all significant variables identified from Phase 1 as well as all the demographic and clinical information, including age, sex, primary payer, BMI, HbA1c, type of residence, cardiovascular disease, chronic kidney disease, use of insulin and non-insulin antidiabetic medications to estimate the effect sizes. Adjusted odds ratios (aOR) and 95% confidence intervals (CI) were reported. 

For the key contextual-level SDoH identified using the two-phase approach, we dichotomized them using the 80th percentile from the key variables as the cutoff. A Higher numeric value in NDI and percent of vacant addresses indicates a neighborhood that is more disadvantaged in socioeconomic profile and has a larger vacancy in addresses. Therefore, we defined neighborhoods with the top 20th percentile in NDI as more deprived neighborhoods, and neighborhoods with the top 20th percentile in percent of vacant addresses as neighborhoods with more occupancy. We applied multilevel logistic regression and adjusted for demographic and clinical characteristics to determine the effect variation by race-ethnicity of key contextual-level SDoH in association with newer ADD initiation.

Analyses were performed using the R statistical software (version 3.6.1; R Development Core Team) and SAS 9.4 (Cary, North Carolina). The study was approved by the Institutional Review Board at the University of Florida (IRB202102283).

## 3. Results

### 3.1. Descriptive Analysis 

Our final analysis comprised 28,874 patients in the cohort. Table 2 highlights the demographic and clinical characteristics of the study population by race and ethnicity. Overall, the mean age was 58 (±15) years, and 61% were women. The majority of the patients were enrolled in public insurance programs such as Medicare (37%) and Medicaid (35%). Compared with NHW patients, NHB patients were younger (54.6 vs. 58.5 years, *p* < 0.01) and more likely to be covered by Medicaid (41% vs. 28%, *p* < 0.01), while Hispanics and patients of other races were older (mean age of Hispanics: 61 years, other race/ethnicity: 60 years), and more likely to be women. Of our cohort, 11,649 patients (40%) had initiated the newer ADD (i.e., SGLT2i or GLP1a). NHW and patients of other races/ethnicities were more likely to have initiated a newer ADD versus another second-line ADD compared to NHB (NHW and other race/ethnicity: both 44%, NHB: 38%, Hispanics: 35%, *p* < 0.01)

### 3.2. Selection of Contextual-Level SDoH

Figure 1 is a volcano plot summarizing the results from Phase 1. After accounting for multiple comparisons using the Benjamin Hochberg procedure, a total of 20 and 11 variables were significantly associated with novel ADD use in the discovery and replication sets, respectively. Among them, ten variables from three categories were significant in both the discovery and replication sets, including the NDI, percentage of low food access (percentage without vehicle access living a half-mile from supply, a food access measure variable), and eight variables documenting the vacant housing in the neighborhood. All ten variables were associated with a lower likelihood of initiating newer ADD (with OR < 1, Figure 1). We observed high correlations among the eight variables documenting vacant land measures (all pairwise correlation coefficients > 0.6, Appendix A, Figure A1). Therefore, we kept only one variable, the percent of vacant addresses in the Phase 2 analysis, as this variable is a more comprehensive measure than the others in the category. 

In Phase 2 analysis, the NDI, percentage without vehicle access living a half mile from supply, and percent of vacant addresses, were simultaneously included in a multivariable logistic regression model after adjusting for baseline demographic and clinical information. Two variables—NDI and percent of vacant addresses—remained statistically significant in the multivariable model. Therefore, our two-phase approach identified two contextual-level SDoH that were significantly associated with a lower likelihood of newer ADD initiation, which are neighborhoods with a higher degree of deprivation and neighborhoods with more vacant housing (Table 3).

### 3.3. Association of Contextual-Level SDoH and New ADD Initiation across Racial and Ethnic Groups

Table 4 shows the results from multivariable logistic regression of binary key contextual-level SDoH variables in association with the novel ADD initiation in the overall cohort and in each racial-ethnic subgroup. In the overall cohort, NHB were significantly less likely to use newer ADD than NHW (aOR 0.82, 95% CI: 0.76–0.88, *p* < 0.01) after adjusting for all the covariates listed above. Patients living in a more deprived neighborhood were associated with a significantly lower likelihood of initiating a newer ADD than the remaining patients (aOR 0.87, 95% CI: 0.81–0.94, *p* < 0.01). Patients living in a neighborhood with more occupancy were less likely to initiate a newer ADD (aOR: 0.91, 95% CI: 0.87–0.95, *p* < 0.01) than their counterparts. We observed similar trends in racial and ethnic subgroups, and no significant interaction of race/ethnicity and contextual-level SDoH was detected. 

## 4. Discussion

SDoH are not only experienced by individuals but also exert influence at the community level. Community-level information about the neighborhoods in which individuals live, learn, work, and play is recognized as the community’s vital signs [18], conveying contextual-level social deprivation and impacting health risks. Our study is unique in linking a set of contextual-level factors documenting social and built environments to extensive collections of EHR data via individuals’ residential histories in a cohort of real-world patients with T2D. Using a data-driven approach, we determined the key contextual-level SDoH factors associated with evidence-based treatment for T2D. After accounting for multiple testing and high correlations among the exposures, two contextual-level SDoH variables characterizing the neighborhood deprivation and vacant housing were identified as being significantly associated with individuals’ limited initiation of newer ADD known to improve cardiorenal outcomes of T2D. These results provide evidence supporting a spatially explicit data-driven approach in developing interventions to address disparities in initiation of T2D treatment.

Increasing evidence has demonstrated an association between neighborhood factors and diabetes outcomes. For example, a more disadvantaged socioeconomic status, poorer food access and built environment (e.g., walkability, recreational facilities), and less social cohesion are associated with the risk of T2D [34,35,36,37]. Additionally, lower neighborhood socioeconomic status was significantly associated with worsening physical and mental health status and poor glycemic control among patients with diabetes [38,39]. However, very few studies have examined whether contextual-level SDoH may influence healthcare quality, such as the initiation of evidence-based treatment. A study conducted using claims data found that contextual-level SDoH, such as poor food access, weak social support, and lack of a healthy built environment, were significantly associated with non-adherence to antihypertensive medication [40]. A randomized trial that enrolled 749 Mexican–American patients at a university-affiliated clinic showed that patients who lived in neighborhoods with greater deprivation were much less likely to adhere to their ADD protocols than those living in neighborhoods in the next higher quartile on the deprivation index [41]. In a US-based study examining the association between neighborhood social environment factors and adherence to oral antidiabetic medications, residents living in neighborhoods with high sociability were more likely to adhere to ADD regimens than their counterparts in less sociable surroundings [42]. 

The current study found that the NDI, an index documenting neighborhood deprivation, was significantly associated with newer ADD initiation. NDI is a composite indicator of contextual-level socioeconomic disadvantages in four areas beyond the strictly specified healthcare setting: income, housing quality, employment, and education. Previous studies have documented the association between neighborhood deprivation, attributed to income, employment, and education, and the quality of diabetes care, reporting that patients living in more deprived neighborhoods were significantly less likely to obtain high-quality diabetes care [7,8,10]. At an individual level, a lack of income and unemployment can create barriers to accessing high-quality diabetes care, while a lack of education has been linked to poor health literacy [43]. At the contextual level, the role of political context could also shape socioeconomic factors, and this interplay could result in unequal resource distribution and structural inequalities in the neighborhoods that perpetuate health disparities [44]. Therefore, individuals with a low socioeconomic profile at the contextual level may face barriers to the use of novel ADD treatment.

The consequences of vacant housing can extend far beyond just an empty space. Vacant land usually is an indicator of population out-migration and disinvestment. In addition to the increased risk of violence and crime [45], vacant housing often leads to a reduction in business and employment, therefore resulting in a lack of community resources, as well as access to essential facilities such as food, medical and social support services [46,47], further exacerbating health disparities in these communities. This lack of resources can have far-reaching consequences on the health and well-being of individuals residing in these areas. Previous studies have shown that empty lots are associated with higher levels of chronic stress and fewer social interactions, and thus resulting in unfavorable health outcomes [25]. In our study, individuals living in a neighborhood with more vacant addresses had lower access to the newer ADD, which could be explained by the lack of access to high-quality diabetes care. It is essential to address the issue of vacant housing and provide necessary resources and support to such disadvantaged communities. Developing innovative strategies, such as mobile medical clinics, have been effective in serving the requirements of medically vulnerable populations, such as the urban poor [48] and populations without stable housing [49], for whom accessibility to fixed healthcare is limited due to the lack of facilities and meager financial resources. MMCs could improve access to care by overcoming geographic and social restrictions, such as neighborhoods with many vacant addresses, which traditional, permanent healthcare facilities must avoid, thus addressing health inequities and mitigating social obstacles to healthcare.

Despite having a disproportionately higher risk of cardiovascular disease, patients from racial and ethnic minority groups have a lower probability of initiating guideline-based therapies that improve their outcomes, including the uptake of new ADD [9,50]. It is suggested that differences might be driven by the disadvantages in insurance coverage and poor socioeconomic status among these racial and ethnic subgroups, and it has been acknowledged in several studies that Medicare Advantage enrollees are less likely to initiate newer ADD than commercial insured patients [7,8,9]. However, in our study, the racial and ethnic disparities in new ADD use persisted after adjusting not only for insurance, but also for NDI, a proxy to socioeconomic status. This represents that such disparity was not driven solely by insurance factors and socioeconomic status. However, we did not identify a significant interaction between race/ethnicity and key contextual-level SDoH in association with initiating newer ADD. While it is possible that the interaction lies elsewhere and was not captured using the two-phase method presented in this study, our findings highlight the structural–environmental factors that drive inequities in the use of evidence-based treatment, independent of race and ethnicity. 

Our study has several limitations. First, our study does not exclude patients with gestational diabetes, and there is a possibility of misclassification for individuals with pregnancy and gestational diabetes but not diagnosed by physicians. Second, the two-phase approach we used did not consider non-linear associations and potential interactions. Generalize additive model could be considered in future work to account for the non-linear relationships among key contextual-level SDoH in association with the study outcome. Additionally, Bayesian kernel machine regression and Bayesian multiple index models can capture the complex interrelations among contextual-level SDoH. Third, although many contextual-level SDoH have been included to characterize the social and built environment, this list is not exhaustive. Continuing efforts are needed to improve the measurement of the contextual-level SDoH further. Fourth, our study cohort was constructed using EHR data, and we cannot completely preclude the prevalence of users of second-line ADD. However, we extended our baseline to three years and restricted individuals who had at least two encounters per year to capture prescription and medical information, which largely eliminated the cases of prevalent users. In addition, regarding the association between individual-level factors and newer ADD initiation, our results were consistent with prior studies using claims data [50], suggesting the validity of the current study’s findings. Finally, participants included in this study were limited to those who received care at one or more sites included in the OneFlorida+ Clinical Research Network. Thus, our results may not be generalizable to those who did not receive healthcare at one of these facilities. 

## 5. Conclusions

In a cohort of T2D patients from a statewide network of EHR, we identified two key contextual-level SDoH factors associated with limited use of new ADD: individuals living in neighborhoods with a higher deprivation index and more vacant addresses were less likely to initiate newer ADD compared with those living in less deprived and more fully occupied neighborhoods. Although the specific mechanisms underlying these associations require further investigation, our findings have contributed to the growing body of evidence of the neighborhood-level factors, their interplay with race across various spatial contexts, and their circumstances on evidence-based healthcare. It is crucial to gain a comprehensive understanding of these complex factors to develop effective strategies for addressing health equities and promoting evidence-based treatment in T2D care. 

## Figures and Tables

**Figure 1 ijerph-20-04036-f001:**
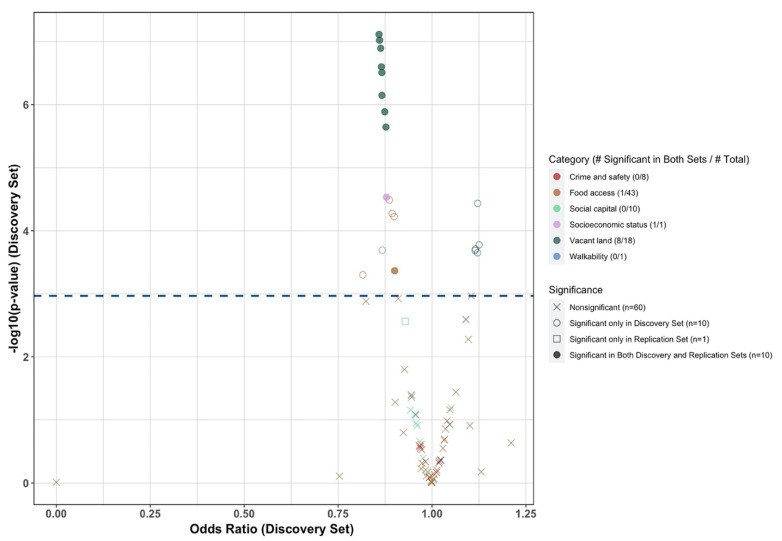
Volcano plot of the results from phase 1 of the two-phase approach.

**Table 1 ijerph-20-04036-t001:** Summary of contextual-level SDoH measures.

Category	Data Source	Time Period	Spatial Scale	Temporal Scale	Number of Variables
Food access	Food Access Research Atlas, USDA	2015, 2019	Census tract	1-year	43
Walkability	Walkability Index, EPA	2006–2013	Census block group	Cross-sectional	1
Vacant land	Aggregated USPS Administrative Data on Address Vacancies, HUD	2015–2019	Census tract	3-month	18
Socioeconomic status	Neighborhood Deprivation Index, ACS	2015–2019	Census block group	5-year	1
Social capital	Census Business Pattern	2015–2019	ZCTA5	1-year	10
Crime and safety	Uniform Crime Reporting Program, FBI	2015–2019	County	1-year	8

Abbreviations: ACS: American Community Survey; EPA: Environmental Protection Agency; USDA, US Department of Agriculture; HUD, Department of Housing and Urban Development; FBI: Federal Bureau of Investigation.

**Table 2 ijerph-20-04036-t002:** Patient Characteristics, by race/ethnicity.

	Overall (n = 28,874)	NHW (n = 11,892)	NHB (n = 10,427)	Hispanics (n = 5458)	Others (n = 818)	*p*-Value
**Age, mean (SD)**	57.65 (15.19)	58.65 (14.44)	54.60 (14.96)	60.84 (16.17)	60.40 (15.06)	<0.0001
**Age group, %(n)**						<0.0001
<25	2.01 (580)	1.54 (183)	2.61 (272)	1.89 (103)	1.71 (14)	
25–34	5.69 (1644)	4.40 (523)	8.00 (834)	4.53 (247)	3.06 (25)	
35–44	11.81 (3410)	10.43 (1240)	14.75 (1538)	9.60 (524)	9.66 (79)	
45–54	19.65 (5674)	19.44 (2312)	21.24 (2215)	17.09 (933)	20.66 (169)	
55–64	28.33 (8180)	29.57 (3516)	28.93 (3017)	25.30 (1381)	22.62 (185)	
65–74	18.68 (5393)	21.21 (2522)	15.14 (1579)	19.13 (1044)	22.62 (185)	
≥75	13.83 (3993)	13.42 (1596)	9.32 (972)	22.46 (1226)	19.68 (161)	
**Female, %(n)**	61.18 (17,666)	55.26 (6571)	67.90 (7080)	62.48 (3410)	55.38 (453)	<0.0001
**Primary payer, %(n)**						<0.0001
Medicare	37.57 (10,847)	38.73 (4606)	33.36 (3478)	44.10 (2407)	31.66 (259)	
Medicaid	34.94 (10,090)	28.85 (3431)	41.44 (4321)	37.38 (2040)	25.43 (208)	
Private insurance	19.96 (5764)	23.74 (2823)	18.64 (1944)	12.42 (678)	32.52 (266)	
No insurance	2.26 (652)	1.95 (232)	3.19 (333)	1.14 (62)	2.81 (23)	
Others	5.27 (1521)	6.73 (800)	3.37 (351)	4.97 (271)	7.58 (62)	
**BMI categories, %(n)**						<0.0001
≤25	9.51 (2745)	9.41 (1119)	9.03 (942)	8.94 (488)	20.78 (170)	
25–30	18.16 (5244)	18.05 (2146)	16.74 (1746)	19.77 (1079)	27.87 (228)	
30–100	54.37 (15,699)	57.69 (6861)	58.22 (6071)	42.84 (2338)	36.92 (302)	
missing	17.96 (5186)	14.85 (1766)	16.00 (1668)	28.45 (1553)	14.43 (118)	
**HbA1c categories, %(n)**						<0.0001
≤7 mmHg	18.43 (5322)	21.72 (2583)	19.37 (2020)	8.67 (473)	25.18 (206)	
7–10 mmHg	23.45 (6772)	28.33 (3369)	23.60 (2461)	10.90 (595)	32.03 (262)	
10–21 mmHg	10.44 (3014)	9.59 (1141)	13.91 (1450)	5.79 (316)	10.27 (84)	
missing	47.68 (13,766)	40.35 (4799)	43.12 (4496)	74.64 (4074)	32.52 (266)	
**Type of residence, %(n)**						<0.0001
Metro Areas	91.99 (26,553)	86.65 (10,300)	94.66 (9868)	97.97 (5345)	95.96 (784)	
Urban or suburban Areas	7.79 (2248)	12.91 (1535)	5.29 (551)	2.00 (109)	3.79 (31)	
Rural Areas	0.22 (63)	0.44 (52)	0.06 (6)	0.04 (2)	0.24 (2)	
**OneFlorida network site, %(n)**						<0.0001
A	17.76 (5127)	18.31 (2177)	14.17 (1478)	23.10 (1261)	20.90 (171)	
B	0.27 (78)	0.45 (53)	0.19 (20)	0	0.12 (1)	
C	3.88 (1121)	3.14 (374)	4.99 (520)	2.91 (159)	7.70 (63)	
D	5.94 (1716)	6.23 (741)	8.47 (883)	0.77 (42)	2.57 (21)	
E	52.11 (15,045)	64.96 (7725)	56.09 (5849)	14.40 (786)	61.25 (501)	
F	20.04 (5787)	6.91 (822)	16.08 (1677)	58.81 (3210)	7.46 (61)	
**CVD, %(n)**	41.26 (11,797)	40.87 (4860)	41.09 (4824)	43.64 (2382)	33.13 (271)	<0.0001
**CKD, %(n)**	33.24 (33.24)	30.21 (3592)	36.55 (3811)	33.86 (1848)	31.17 (255)	<0.0001
**Insulin use, %(n)**	48.59 (14,029)	45.63 (5426)	53.28 (5556)	47.45 (2590)	41.20 (337)	<0.0001
**Any non-insulin antidiabetics use, %(n)**	67.30 (19,432)	66.25 (7878)	67.33 (7021)	69.02 (3767)	70.29 (575)	0.0008
**Metformin, %(n)**	45.68 (13,190)	45.01 (5353)	45.98 (4794)	46.01 (2511)	48.78 (399)	0.117
**DPP-4 inhibitors, %(n)**	21.27 (6141)	21.41 (2546)	19.62 (2046)	23.43 (1279)	23.59 (193)	<0.0001
**Sulfonylureas, %(n)**	31.49 (9092)	29.09 (3459)	31.77 (3313)	35.87 (1958)	35.21 (288)	<0.0001
**Thiazolidinediones, %(n)**	4.68 (1351)	5.05 (601)	3.86 (403)	5.15 (281)	6.60 (54)	<0.0001

Abbreviations: SD, standard deviation; BMI, body mass index; HbA1c, hemoglobin A1c; CVD, cardiovascular disease; CKD, chronic kidney disease; DPP-4, dipeptidyl-peptidase 4. Values are means (SD) for continuous variables; percentages or ns or both for categorical variables. Values of polytomous variables may not sum to 100% due to rounding.

**Table 3 ijerph-20-04036-t003:** Significant social determinants of health variables associated with the use newer antidiabetic drugs identified by the two-phase approach.

Exposure		Transformation	Standard Deviation	Phase 1				Phase 2	
				Discovery Set		Replication Set			
Variable	Category			OR (95% CI)	q-Value	OR (95% CI)	q-Value	OR (95% CI)	*p*-Value
Percentage of low access population with housing units without vehicle access at 1/2 mile	Food access	log_x	0.165	0.90 (0.85, 0.95)	0.0349	0.86 (0.81, 0.91)	<0.0001	0.96 (0.91, 1.01)	0.0905
Neighborhood deprivation index	Socioeconomic status	Yeo-Johnson (lamda = −0.49)	2.052	0.88 (0.83, 0.93)	0.0024	0.84 (0.79, 0.89)	<0.0001	0.92 (0.88, 0.97)	0.0031
Percent of vacant addresses	Vacant land	Square Root	0.089	0.86 (0.81, 0.91)	<0.0001	0.86 (0.82, 0.91)	<0.0001	0.91 (0.87, 0.95)	<0.0001

Abbreviations: OR, odds ratio; CI, confidence interval. Odds ratio (OR) and 95% confidence interval (CI) for each standard deviation increase.

**Table 4 ijerph-20-04036-t004:** Results from multivariate logistic regressions (with binary contextual-level social determinants of health).

Variable		Overall (n = 28,874)		NHW Subgroup (n = 11,892)		NHB Subgroup (n = 10,427)		Hispanic Subgroup (n = 5458)		Other Race Subgroup (n = 818)	
	Category	aOR (95% CI)	*p*-Value	aOR (95% CI)	*p*-Value	aOR (95% CI)	*p*-Value	aOR (95% CI)	*p*-Value	aOR (95% CI)	*p*-Value
Neighborhood deprivation index	Socioeconomic Status	0.87 (0.81, 0.94)	0.0003	0.80 (0.67, 0.95)	0.0121	0.91 (0.82, 1.00)	0.0517	0.83 (0.70, 0.97)	0.0215	0.81 (0.40, 1.66)	0.5653
Percent of vacant addresses	Vacant Land	0.91 (0.85, 0.98)	0.0087	0.95 (0.84, 1.08)	0.46	0.87 (0.79, 0.96)	0.0073	0.96 (0.77, 1.18)	0.6899	0.92 (0.54, 1.57)	0.7645

Abbreviations: aOR, adjusted odds ratio; CI, confidence interval; NHW, non-Hispanic White; NHB, non-Hispanic Black. Adjusted odds ratio (aOR) and 95% confidence interval (CI) for comparing the top 20th percentile to bottom 80th percentile (for neighborhood deprivation index: comparing more deprived neighborhoods compare to less deprived neighborhoods; for percent of vacant addresses: comparing more occupancy neighborhoods compare to less occupancy neighborhoods) while adjusting for age, sex, primary payer, BMI, HbA1c, type of residence, cardiovascular disease, chronic kidney disease, use of insulin and non-insulin antidiabetic medications.

## Data Availability

The data presented in this study are available on request from the corresponding author. The data are not publicly available due to privacy restrictions.

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
