# Peer review of "Impact of Contextual-Level Social Determinants of Health on Newer Antidiabetic Drug Adoption in Patients with Type 2 Diabetes"

_ijerph, 2023, doi:10.3390/ijerph20054036_

Round 1
Reviewer 1 Report
Authors discuss their study title 'Impact of contextual-level social determinants of health on 2 newer antidiabetic drug adoption in patients with type 2 3 diabetes'. While I found this study useful for the readers. However I have two main concerns regarding the study.
First, the cohort construction criteria to identify T2D patients was not clear to me. Specifically, I am not sure if authors removed patients with gestational diabetes before the index date. If not, then the gestational diabetes may bias the inference and must be accounted for. Additionally, what is the back ground distribution of number of clinical visits in the past three years before the index date? It is very hard to relate the length of patient record in the absence of such distribution. For example, what if a patient had two clinical visits just two days before the index date? If that is the case then such patients likely to bias the estimates.
Second, I feel that a comprehensive analysis is needed on imputing the SDoH (or other demographics) that are missing from the data. It is now understood and well accepted that SDoH and other demographics are inherently biased, therefore, does imputing those missing values likely to amplify the bias? It'll be helpfull if authors could provide some sort of empirical evidence on the overall distribution of the SDoH and other demographics with and without imputation to better able to put the results in context.
Reviewer 2 Report
Thank you for the opportunity to review this interesting manuscript. It describes an observational study which aims to investigate associations between contextual social determinants of health (SDoH) and the adoption of novel antidiabetic drugs (ADDs). While the analysis is at the patient level, it primarily makes use of data on SDoH collected at the neighborhood level.
In the end, the authors find that two contextual SDoH variables are significantly associated with novel ADD initiation: NDI and the percentage of vacant housing at the neighborhood level. Both were associated with a lower likelihood of novel ADD initation.
Altogether, this is a solid piece of work that has had a lot of thought put into it. My major and minor comments for the authors are below.
Major comments
- The Phase 1 analysis uses the B-H method (as per the results) to control FDR but this is not described in section 2.4, “Statistical analysis”. Recommend more completely describing the approach in 2.4 and not backloading to the Results. How was the correlation threshold of 0.6 chosen? Seems a little arbitrary.
- The current two-phase analysis, where variable selection is performed before fitting the outcome models, seems potentially conservative for this sort of study. Recommend fitting a multilevel model in one step, at least as a sensitivity analysis. See Gelman et al., "Why We (Usually) Don’t Have to Worry About Multiple Comparisons.”
- The finding that 8/10 variables were those that recorded the level of vacant housing in the neighborhood is an interesting one, and one that I think is (unintentionally) glossed over by the authors. Insofar as vacant housing may be related to separation from services and high-resource neighborhoods, these variables could be acting as proxies for access. This is briefly discussed in the Discussion section (page 9, line ~290) but is something that I think deserves more focus. Would it be possible to correlate or otherwise account for proximity to services in your analysis?
Minor comments
- Why were the 2 SDoH dichotomized? Guidelines generally advise against dichotomizing continuous variables. Would be interesting (and would recommend) to apply a GAM as a sensitivity analysis to understand more fully the relationship between the SDoH and ADD initiation outcomes over the ranges of the SDoH variables, and to support the decision to dichotomize.
- One thing to consider is that the subjects in the more deprived neighborhoods are exceptional in a sense that they managed to enter the dataset. But what about the other individuals in these same neighborhoods that weren’t captured by this analysis? One possibility is that they may have had poorer outcomes in terms of ADD initiation. But also equally plausible is that they may have been served by other programs that don’t show up in these data. As a potential limitation, I think it would be helpful to at least get a sense of the potential scope of this issue
- Figure A1. Would be good to have meaningful variable labels in this figure. Also consider reporting only the upper or lower triangular values in the correlation matrix.
- Page 3, line 128-129. NACIS should be NAICS.
Reviewer 3 Report
The authors use a large, multi-state EMR database to test the association between "contextual-level SDoH" and the use of diabetic medications which have a rapidly mounting base of evidence to support their merits as superior treatments for diabetes. I was excited to read this paper because I agree it is a crucial frontier in achieving health equity in diabetes outcomes. However, I have substantial concerns that the authors do not present the information adequately and it lacks a scholarly understanding of SDoH and structural drivers of inequality overall. As written, the paper does not add much to the literature because it does not draw any meaningful conclusions from its findings.
Some overarching concerns:
1) Introduction - does not build an adequate case for why SGLT-2 and GLP1 are considered important in diabetes treatment. Renal and Cardiac protection are important, as are significant effect size in A1C, weight loss, and limited risk of hypoglycemia (which is why they are preferred to insulin and sulfonyureas). These reasonings are all explained in ADA guidance.
2) Further - the introduction as written does not provide the reader which much of understanding of what contextual-level social determinants of health are and why they are important. I am a health equity researcher and I was not sure, so the average researcher will not be able to understand the case being built in this introduction . Are they neighborhood factors? Are they the built environment? How should the reader understand them in terms of their impact on diabetes outcomes? Are they considered a potential intervention point? Some of the detail in the first paragraph of the discussion may be better suited in the introduction, and it should go further than that. The reader needs to understand what the c-l SDoH are, what the literature shows about them and diabetes outcomes, and how understanding this gap in research could lead to positive changes. As written none of this is clear. For example "Previous studies have demonstrated the association of contextual level SDoH with geographic variation in diabetes risk" is vague and meaningless.
2) Methods - the timing of this data collection is important. These drugs has just achieved market status in 2015 and gained momentum in prescriber comfort (and guideline support) during the study period. For example, I do not believe they were recognized as second-line drugs by the ADA until 2018. Do you include an indicator for time in your models?
"All of the potential confounders" is not an adequate description of which covariates were included in the models.
Page 8 - "We dichotomized two key contextual-level SDoH..." - this sentences does not make sense but I believe you are saying you dichotomized your independent variable. How did you choose your cut point? are the top 20% the most advantaged or least advantaged? Consider revising. Does this belong in the Methods section?
Results - I find the results difficult to interpret and understand.
Discussion - the role of structural inequalities should be included in the conversation about NDI. Positing CHWs as a silver bullet solution is not plausible. Consider including how your findings fit into the larger conversation of "pharamcoequity" as described in multiple papers by Utibe Essien, et al.
